# Non-*O* ABO blood group genotypes differ in their associations with *Plasmodium falciparum* rosetting and severe malaria

D. Herbert Opi [1,2¤]*, Carolyne M. Ndila[1], Sophie Uyoga[1], Alex W. Macharia[1], Clare Fennell[2], Lucy B. Ochola[1], Gideon Nyutu[1], Bethseba R. Siddondo[1], John Ojal[1], Mohammed Shebe[1], Kennedy O. Awuondo[1], Neema Mturi[1], Norbert Peshu[1], Benjamin Tsofa[1], Gavin Band[3], Kathryn Maitland[1,4], Dominic P. Kwiatkowski [3†], Kirk A. Rockett[3], Thomas N. Williams[1,4☯], J. Alexandra Rowe[2☯]

1 Kenya Medical Research Institute-Wellcome Trust Research Programme, Kilifi, Kenya, 2 Centre for Immunity, Infection and Evolution, Institute of Immunology and Infection Research, School of Biological Sciences, University of Edinburgh, Edinburgh, United Kingdom, 3 Wellcome Centre for Human Genetics, Oxford, United Kingdom, 4 Institute for Global Health Innovation, Department of Surgery and Cancer, Imperial College, London, United Kingdom

☯ These authors contributed equally to this work.
† Deceased.
¤ Current address: Burnet Institute, Melbourne, Victoria, Australia
* herbert.opi@burnet.edu.au

**Data Availability Statement:** De-identified participant data, in vitro datasets and numerical data for all the figures and summary statistics used during the current study are available at https://doi.

## Abstract

Blood group O is associated with protection against severe malaria and reduced size and stability of *P. falciparum*-host red blood cell (RBC) rosettes compared to non-O blood groups. Whether the non-O blood groups encoded by the specific *ABO* genotypes *AO*, *BO*, *AA*, *BB* and *AB* differ in their associations with severe malaria and rosetting is unknown. The A and B antigens are host RBC receptors for rosetting, hence we hypothesized that the higher levels of A and/or B antigen on RBCs from *AA*, *BB* and *AB* genotypes compared to *AO/BO* genotypes could lead to larger rosettes, increased microvascular obstruction and higher risk of malaria pathology. We used a case-control study of Kenyan children and *in vitro* adhesion assays to test the hypothesis that "double dose" non-*O* genotypes (*AA*, *BB*, *AB*) are associated with increased risk of severe malaria and larger rosettes than "single dose" heterozygotes (*AO*, *BO*). In the case-control study, compared to *OO*, the double dose genotypes consistently had higher odds ratios (OR) for severe malaria than single dose genotypes, with *AB* (OR 1.93) and *AO* (OR 1.27) showing most marked difference (*p* = 0.02, Wald test). *In vitro* experiments with blood group A-preferring *P. falciparum* parasites showed that significantly larger rosettes were formed with *AA* and *AB* host RBCs compared to *OO*, whereas *AO* and *BO* genotypes rosettes were indistinguishable from *OO*. Overall, the data show that *ABO* genotype influences *P. falciparum* rosetting and support the hypothesis that double dose non-*O* genotypes confer a greater risk of severe malaria than *AO/BO* heterozygosity.

org/10.7910/DVN/A5KILM and can be accessed on request from the KEMRI-Wellcome Trust Research Programme Data Governance Committee (dgc@kemri-wellcome.org).

**Funding:** This work was supported by the Wellcome Trust through Senior Research Fellowships (grant number 084226 to JAR) and (grant numbers 091758 and 202800 to TNW), through core support to the KEMRI-Wellcome Trust Research Programme (grant number 203077 to TNW) and through a sub-grant from a Wellcome Trust Strategic Award (grant number 084538 to DHO). The work was partially funded by the European Community's Seventh Framework Programme (FP7/2007-2013) (grant agreement 242095 to TNW) and by the UK Medical Research Council (grant number G0600718 to TNW). The MalariaGEN Consortium was supported by the Wellcome Trust (grant number 077383 to DPK) and by the Foundation for the National Institutes of Health (grant number 566 to DPK) as part of the Bill & Melinda Gates Grand Challenges in Global Health Initiative. The Resource Centre for Genomic Epidemiology of Malaria was supported by the Wellcome Trust (grant number 090770 to DPK). The Wellcome Trust also provided core awards to the Wellcome Trust Centre for Human Genetics (grant number 090532 to DPK) and to the Wellcome Trust Sanger Institute (grant number 098051 to DPK). This work was also supported by the Medical Research Council (grant number G19/9 to DPK). The funders had no role in study design, collection, analysis and interpretation of data, in the writing of the manuscript or in the decision to submit the paper for publication.

**Competing interests:** The authors have declared that no competing interests exist.

## Author summary

The most common human blood group, ABO, affects susceptibility to multiple diseases including malaria, whereby the non-O blood groups A, B and AB are associated with an increased risk of severe malaria in comparison to blood group O. This may occur because red blood cell (RBC) surface-expressed non-O blood group antigens mediate binding to *P. falciparum*-infected RBCs to form clumps known as rosettes, that occlude microvasculature circulation and cause severe malaria pathology. Nevertheless, to date, these conclusions have largely been based on ABO blood groups determined by serological antibody typing. Genetic classification into *AO*, *AA*, *BO*, *BB*, *AB* and *OO* offers a finer and more specific classification of ABO blood groups, but associations with severe malaria and rosetting based on this method have not been described previously. In a case-control study of >5000 Kenyan children, we show that the "double dose" non-*O* genotypes *AA*, *BB* and *AB* are associated with an increased risk to severe malaria compared to the "single dose" non-*O* genotypes *AO* and *BO*, and that this is most significant for *AB* versus *AO*. In *in vitro* experiments, double dose non-*O* genotypes formed larger rosettes compared to single dose non-*O* genotypes, providing a potential explanation for their increased severe malaria risk.

## Introduction

The ABO blood group was the first human blood group system to be discovered, and has since been widely studied [1]. Three red blood cell (RBC) expressed carbohydrate antigens characterise the ABO system; the A and B antigens are the products of the enzymatic addition of N-acetyl-D-galactosamine (GalNAc) or D-galactose (Gal), respectively, to a precursor H antigen, while an inactive glycosyltransferase fails to add any sugar residues to the H antigen giving rise to blood group O [2]. The resultant A, B, AB and O blood group phenotypes have been associated with numerous diseases [3], most recently including COVID-19 [4], and there is strong evidence to support their role in severe malaria [5]. Blood group O has a high frequency in malaria-endemic regions [6], and has been associated with protection from severe malaria in numerous studies [7–15]. Both observations are consistent with a malaria-selective pressure for blood group O. Additionally, the A and B antigens interact with molecules on the surface of *Plasmodium falciparum* infected red blood cells (iRBCs), such as *P. falciparum* erythrocyte membrane protein 1 (PfEMP1), mediating iRBC binding to uninfected RBCs to form clusters called rosettes [16–20]. Other *P. falciparum* iRBC surface expressed parasite ligands implicated in rosetting include the repetitive interspersed family proteins (RIFIN) interacting with blood group A [19] and the sub-telomeric variant open reading frame (STEVOR) proteins [21]. *P. falciparum* isolates vary in their propensity to form rosettes, high levels of rosetting contributing to impaired microvascular blood flow, and severe malaria [22–24]. Rosetting iRBCs show a "preference" for uninfected A or B RBCs, and form larger, more stable rosettes than with O RBCs [16–19,25–27]. The protection against severe malaria associated with blood group O may be due to reduced rosetting, and hence reduced microvascular obstruction and pathology [7,16,25,28–30].

Previous studies linking ABO blood group to malaria susceptibility and rosetting have either used serological RBC agglutination assays to determine ABO blood group phenotype (A, B, AB, O), or single nucleotide polymorphisms (SNPs) at the *ABO* gene to derive *ABO* genotype (*OO, AO, AA, BO, BB, AB*), which have then been used to infer ABO phenotypes. While multiple polymorphisms underlie the genetic basis of the ABO system [31,32], most are

rare, and two SNPs rs8176719 and rs8176746 are considered sufficient (>90% accuracy) to infer ABO phenotype [8–15,33,34]. However, the concordance between *ABO* genotype and ABO phenotype has not been reported in previous population studies of malaria susceptibility, most of which have been conducted in sub-Saharan Africa.

Similarly, few studies have been conducted which have examined the association between *ABO* genotype (rather than phenotype) and malaria susceptibility, or potential protective mechanisms such as their impacts on rosetting. Prior work shows a gene dosage effect between *ABO* genotype and the number of ABH antigens on RBCs. For example, *AA* and *BB* homozygotes have substantially higher levels of A and B antigens on RBCs than *AO* and *BO* heterozygotes respectively [35,36], while *AB* heterozygotes have antigen levels that are more similar to those found in *AA/BB* homozygotes [35–39]. The level of A antigen displayed on the RBC surface has been shown to have an impact on rosetting. For example, both rosetting and PfEMP1-binding are substantially reduced in RBCs of the $A_2$ blood group phenotype, in which fewer A antigen sites are displayed per RBC than cells of the common $A_1$ phenotype [16,17,40]. However, to the best of our knowledge, whether RBCs from *AO* and *BO* genotypes differ in their rosette-forming ability compared to RBCs from *AA/BB/AB* genotypes with higher levels of A and B antigens has not yet been tested.

Here, we used a case-control study conducted in East Africa to investigate whether specific *ABO* genotypes are associated with differing levels of susceptibility to severe childhood malaria. In addition, we examined the concordance between *ABO* genotype and phenotype. We also investigated whether *ABO* genotypes are associated with risk of uncomplicated malaria or asymptomatic infection in a longitudinal cohort and cross-sectional study, respectively. Finally, through *in vitro* studies, we investigated whether *ABO* genotype influences either *P. falciparum* rosetting or other parasite adhesion-related phenomena including binding to endothelial receptors and the display of PfEMP1 on the iRBC surface [41]. We hypothesised that individuals with two non-*O* alleles (*AA/BB/AB*) might have a greater risk of severe malaria than non-*O* heterozygotes (*AO/BO*), due to the former having increased levels of A or B antigens on their RBCs, enabling iRBCs to form larger, more stable rosettes that cause greater microvascular obstruction and pathology.

## Methods

### Ethics and approval and consent to participate

Individual written informed consent was obtained from the parents or guardians of all study participants. All study protocols were approved by the Kenya Medical Research Institute (KEMRI) National Ethical Review Committee (case control study: SCC1192; cohort study: SCC3149). RBCs and sera from Scottish blood donors were obtained following informed consent, with approval from the Scottish National Blood Transfusion Service Committee for the Governance of Blood and Tissue Samples for Nontherapeutic Use (reference SNBTS 12~35).

### Kilifi study area

All epidemiological and clinical studies were conducted in Kilifi County on the coast of Kenya [42]. At the time the studies were conducted, malaria transmission followed a seasonal pattern determined by annual long and short rainy seasons [43].

### Kilifi case-control study

This study has been described in detail elsewhere [14,44,45]. 2245 children aged <14 years who presented to Kilifi County Hospital (KCH) with features of severe malaria were recruited

as cases between January 2001 and January 2008. Severe malaria was classified as the presence of a blood film positive for *P. falciparum* accompanied by any of the following complications: cerebral malaria (CM, a Blantyre Coma Score of <3), severe malarial anaemia (SMA, a haemoglobin concentration of <5g/dl) or respiratory distress (RD, abnormally deep breathing) [46]. In a recent modelling study based on white blood cell and platelet counts in the same cohort of children, it was shown that malaria was probably not the primary cause for the severe disease seen in approximately one third (842 out of 2245) of the "severe malaria" cases [47]. For this reason, these cases were dropped from the current analysis, leaving 1403 cases for inclusion in the current study. Controls (n = 3949) were healthy children who were born within the study area between August 2006 and September 2010 and who were recruited at 3–12 months of age into a genetics cohort study [48]. Therefore, controls were matched to cases by location but differed from cases in age-structure. While not typical of classical case-control design, cord blood or infant samples have been widely used as controls in previous genetic association studies conducted across sub-Saharan Africa [12,13,49], because this provides the most feasible way of collecting large sample numbers in resource-limited settings. Data on ABO blood group (inferred from genotype data) and severe malaria risk from this study have been published previously [14].

## Kilifi longitudinal cohort study

The Kilifi longitudinal cohort study, described in detail elsewhere [50], was established to investigate the immuno-epidemiology of uncomplicated malaria and other common childhood diseases in an area approximately 15 km to the north of KCH [51,52]. Between August 1998 and August 2001, children aged <10 years were recruited into the cohort, including children born from study households during the study period. Cohort members were followed up actively for clinical events on a weekly basis. In addition, members were passively followed for inter-current illnesses at a dedicated clinic at KCH. Finally, the cohort was also monitored for the prevalence of asymptomatic *P. falciparum* carriage through four cross-sectional surveys carried out in March, July and October 2000 and June 2001. Exclusion criteria included death, migration from the study area for more than 2 months and consent withdrawal. Uncomplicated malaria in this cohort was defined as fever (axillary temperature of >37.5˚C) with *P. falciparum* infection at any density, in the absence of any signs of severity [53].

## ABO phenotyping and genotyping

Serological typing for ABO blood group phenotypes was carried out by slide haemagglutination assays using anti-A and anti-B monoclonal antibodies (Alba Bioscience, Edinburgh, UK). For genotyping, DNA was extracted from fresh or frozen whole blood samples using either an ABI PRISM 6100 Nucleic acid prep station (Applied Biosystems, Foster City, CA, USA) or QIAamp DNA Blood Mini Kits (Qiagen, West Sussex, United Kingdom) respectively. *ABO* genotyping was carried out by assessing SNPs rs8176719 and rs8176746, using the SEQUENOM iPLEX Gold (Sequenom) multiplex system following DNA amplification by whole genome amplification, as described previously [8]. rs8176719 in exon 6 of the *ABO* gene on chromosome 9 encodes the 261G deletion giving rise to the *O* allele [31]. rs8176746 in exon 7 of the *ABO* gene encodes C796A which distinguishes A and B alleles [31]. The two SNPs were used to designate *ABO* genotypes and infer blood groups as described previously in this population [8] and as summarized in Table 1. The common *O* deletion (D) arose on the background of the A allele (C). Therefore, double heterozygotes GD X AC are assumed to have the haplotype GA and DC, giving rise to the *BO* genotype, rather than haplotype GC and DA that gives rise to the *AO* genotype (Table 1). Genotyping for rs334 on *HBB*, which detects the HbS

**Table 1. *ABO* genotype and phenotype frequencies in the case-control study.**

| rs8176719 X rs8176746 haplotype[§] | Inferred *ABO* genotype | Cases N (%) | Controls N (%) | ABO serological phenotype[#] | Controls N (%) |
|---|---|---|---|---|---|
| DD X CC | *OO* | 605 (43.3) | 2030 (52.5) | O | 1540 (55.8) |
| DD X AA | *OO* | 3 (0.2) | 1 (0.0) | | |
| DD X AC | *OO* | 14 (1.0) | 87 (2.3) | | |
| GD X CC | *AO* | 306 (21.9) | 810 (21.0) | A | 637 (23.1) |
| GG X CC | *AA* | 37 (2.7) | 83 (2.1) | | |
| GD X AA | *BO* | 13 (0.9) | 20 (0.5) | B | 503 (18.2) |
| GD*X AC | *BO* | 324 (23.2) | 663 (17.2) | | |
| GG X AA | *BB* | 34 (2.4) | 55 (1.4) | | |
| GG X AC | *AB* | 61 (4.4) | 115 (3.0) | AB | 81 (2.9) |

[§]*ABO* blood group genotypes were determined using two SNPs at the *ABO* locus; rs8176719 in exon 6 that encodes the 261G deletion giving rise to blood group O with the rs8176719 alleles represented as 261G (G) and 261delG (D); rs8176746 in exon 7 which encodes C796A and distinguishes *A* and *B* alleles. Genotyping at both SNPs was successful in 99.6% (1397/1403) severe malaria cases and 97.8% (3864/3949) community controls. 1 severe malaria case and 8 controls had missing genotype data for the rs8176746 SNP but were homozygous for the 261G deletion in exon 6 and were therefore denoted as *OO* genotype.

* The common O deletion (D) arose on the background of the A allele (C). Therefore, double heterozygotes GD X AC are assumed to have the haplotype GA and DC, giving rise to the *BO* genotype, rather than haplotype GC and DA that gives rise to the *AO* genotype. This assumption is supported by the data shown here, with only 4/5261 children being homozygous for the D deletion on the background of the B allele (AA).

[#]ABO blood group serological phenotype was determined using standard haemagglutination tests on a sub-group of 2761 (70%) of the community controls. ABO phenotyping was not included as part of the original study design and was only added after the study began, so some controls were not phenotyped.

mutation that results in sickle cell trait (HbAS) or sickle cell anaemia (HbSS), and for the 3.7Kb α-globin deletion at the *HBA* locus, which gives rise to the common African variant of $\alpha^+$thalassaemia, was conducted by PCR as described elsewhere [54,55]. Compatibility with Hardy-Weinberg Equilibrium (HWE) predictions was determined for *ABO* genotype SNPs in control samples [12]. SNPs displaying large departures from HWE among controls ($p<0.05$) can reflect underlying data problems such as genotyping errors or sample bias and should generally be excluded from use in association analyses [56]. In the current study, we found no significant departures from HWE with regard to *ABO* SNPs among controls ($\chi^2 = 0.78$, $p = 0.677$), and as such, all SNPs were included in our analysis.

## Data analysis for epidemiological studies

Agreement between *ABO* blood group genotype and serological phenotype was tested using Cohen's Kappa statistic, for which scores of 1 and 0 signify complete agreement and no agreement, respectively.

The Pearson's $\chi^2$ or Fisher's exact tests were used to test for differences in the distribution of *ABO* genotypes between severe malaria (including its various clinical sub-types of CM, SMA, RD and mortality) and community controls or across groups in different categorical variables including gender, ethnic group and HbS and $\alpha^+$thalassaemia genotypes. We tested for differences in age across *ABO* genotypes by use of the Kruskal Wallis test.

Odds ratios (ORs) (with 95% confidence intervals, CIs) for severe malaria and its various clinical sub-types were determined by comparison of genotype frequencies among cases and controls, using a fixed-effects logistic-regression model. We conducted both a univariate and multivariate analysis with adjustments for self-reported ethnicity, gender, $\alpha^+$thalassaemia and HbAS. The *OO* genotype was taken as the reference group to which the other blood group genotypes (*AO*, *AA*, *AB*, *BO*, *BB*) were compared both individually and as 'non-*O*' genotypes combined. To test the hypothesis that double dose non-*O* genotypes (*AA*, *AB*, *BB*) are

associated with a higher risk of severe malaria than single dose non-*O* genotypes (*AO*, *BO*), ORs between single and double dose non-*O* genotypes from the logistic regression analysis above were compared using the Wald test. As an alternative approach, logistic regression analysis was carried out as described above, but instead comparing the ORs of severe malaria for double dose genotypes *AA* and *AB* to single dose *AO* as the reference genotype, and similarly comparing *BB* and *AB* to *BO* as the reference genotype.

In the longitudinal cohort study, the impact of *ABO* genotype on the incidence of uncomplicated malaria was investigated using a random effects Poisson regression model with the reference *OO* genotype compared to all other *ABO* genotypes assuming a genotypic model of inheritance, or to non-*O* genotypes combined in a recessive model of inheritance using the rs8176719 SNP. Incidence Rate Ratios (IRR) were generated from both a univariate model and a multivariate model which were both adjusted for age, season, ethnic group, and HbS and $\alpha^+$thalassaemia genotypes. All analyses accounted for within-person clustering of events. For the cross-sectional survey, ORs investigating the impact of *ABO* genotypes on the prevalence of asymptomatic parasitaemia were generated by use of logistic regression analysis. We conducted both a univariate analysis and multivariate analysis with adjustment for confounding by age, season, ethnic group and HbS genotype. The analysis also accounted for clustering of asymptomatic parasitaemia events within individuals.

In the epidemiological analyses, adjustments for multiple testing were not performed; instead, all adjusted ORs, confidence intervals and *p* values have been clearly reported and it is emphasized that no single study is conclusive and additional studies are needed to determine if results are replicable. This approach has been suggested previously for epidemiological data, to avoid potentially important findings being discarded (type II error) due to the stringency of multiple comparison adjustments [57–59].

## Red blood cells

All *in vitro* assays were carried out using malaria negative RBC samples that were collected and processed during May 2009 and May 2010, as described in detail previously [41], as part of the Kilifi longitudinal cohort study, [51,52]. Briefly, whole blood samples were collected into heparinized tubes with plasma aspirated and removed following centrifugation, and white blood cells removed by density centrifugation through Lymphoprep (Fresenius Kabi Norge AS for Axis-Shield PoC AS, Oslo, Norway). Purified RBC pellets were washed and either stored at 4˚C and used within 4 days for cytoadhesion assays or cryopreserved in glycerolyte and thawed by standard methods [60, 61] before use for rosetting and PfEMP1 expression assays.

## Parasites and parasite culture

Rosetting assays were carried out with the *P. falciparum* clone IT/R29 [62–66] expressing the ITvar9 PfEMP1 variant that binds to RBCs to form rosettes [62–67]. The ItG *P. falciparum* line used for the *in vitro* static adhesion assays binds to both ICAM-1 and CD36 endothelial receptors and expresses the ITvar16 PfEMP1 variant [62,63]. IT/R29 and ItG are clones derived from the Brazilian IT4/25/5 parasite strain following selection for rosetting (IT/R29) or ICAM-1 binding (ItG) [64]. Due to antigenic switching, both adhesion phenotypes are gradually lost in long-term parasite culture. Therefore, frozen stocks of ItG previously selected for ICAM-1 binding were thawed and used within three weeks [68] while IT/R29 was maintained at a baseline rosetting frequency of >50% by repeated selection using gelatin flotation or density centrifugation on 60% Percoll (Sigma) as previously described [69].

All parasites were maintained using standard culture methods [61] in blood group O + human RBCs. Prior to conducting the *in vitro* experiments, mature pigmented-trophozoite

stage iRBCs were purified to >90% parasitaemia from uninfected and ring-stage iRBCs by magnetic–activated cell sorting (MACS) [70]. The purified iRBCs were then used to invade donor RBCs of different *ABO* genotypes, in duplicate flasks or duplicate wells in 96-well flat-bottomed plates, at a starting parasitaemia of 1.5% for ItG and 3% for IT/R29 and cultured for 24–48 hours to the mature pigmented-trophozoite stage [41]. RBCs from a local control *OO* donor were included in all experiments to adjust for day-to-day variation in the assays.

## Rosetting assays

Rosetting was assessed as described previously [41,61]. Briefly, a wet preparation of IT/R29 culture suspension at 2% haematocrit stained with 25μg/ml ethidium-bromide was examined with combined UV/bright field using a Leica DM 2000 fluorescence microscope (x40 objective). A rosette was defined as a mature pigmented trophozoite-stage iRBC binding two or more uninfected RBC, and the number of uninfected RBC bound per rosette was counted for at least 30 rosettes to determine mean rosette size for each donor. The frequency of large rosettes is the percentage of the rosettes counted that had >4 uninfected RBCs per rosette. Sample genotypes were masked to avoid observer bias.

## Static adhesion assays

Static adhesion assays were carried out as described in detail previously [41,68]. Briefly, donor RBC samples infected with ItG *P. falciparum* parasites were assessed for binding to purified recombinant proteins CD36 (R & D Systems, UK) and ICAM1-Fc (a gift from Professor Alister Craig, Liverpool School of Tropical Medicine) spotted on bacteriological petri dishes (BD Falcon 351007) at a concentration of 50μg/ml. Each donor sample was tested once in duplicate dishes run on the same day, with triplicate spots of each protein in each dish. For each spot, 6 images of adherent iRBCs were captured across random fields using an inverted microscope (Eclipse TE2000-S, Nikon, x40 magnification), giving 36 images per protein for each donor RBC. Images were processed and analysed using Image SXM software (University of Liverpool, UK) [71] and the results expressed as the mean number of iRBC bound per $mm^2$ of surface area.

## PfEMP1 expression assays

Assessment of PfEMP1 expression by flow cytometry, including gating strategies, in donor RBCs infected with the IT/R29 *P. falciparum* clone has been described in detail previously [41]. Briefly, ITvar9 PfEMP1 expression was determined by staining the same preparation of IT/R29 donor iRBC samples tested for rosetting above, with rabbit polyclonal total IgG raised against the ITvar9 variant [72]. ITvar9 PfEMP1 expression was defined by both the median fluorescent intensity (MFI) and proportion of iRBC positively staining with anti-ITvar9 IgG.

## Data analysis for *in vitro* experiments

Multivariate regression analysis was used to test the effect of *ABO* genotype on rosetting, cytoadhesion or PfEMP1 expression. Potential confounding variables including HbS and $\alpha^+$thalassaemia genotypes [41], mean corpuscular volume and complement receptor 1 level [73,74] and Knops blood group [41,44] were first examined in univariate analyses. However, the final model only included the confounding variables HbAS and/or $\alpha^+$thalassaemia which showed significant associations on univariate analysis (p<0.05) and an interaction term between the two, and improved the overall model fit tested using the log-likelihood ratio test. Experimental day was included as a covariate to account for day-to-day variation when

experiments were conducted over several days. Non-normally distributed CD36 and ICAM-1 binding data were normalized by square root-transformation. The final list of variables adjusted for in each analysis is shown in the footnote under each respective results table. To visualise the data, dot plots were generated showing individual data points for each donor, normalized to the mean of control donor genotype *OO* cells run on the same day to account for day-to-day variation. Non-*O* blood group genotypes were compared to the reference *OO* genotype using a Kruskal Wallis test with Dunn's multiple comparisons. A *p* value of <0.05 was considered significant in all analyses. Statistical analyses were performed in Stata v13.1 (Stata-Corp, Texas, USA) or Prism v7.0 (Graphpad Inc, San Diego, California) and graphs were generated using Prism v7.0.

## Blood group preference of the IT/R29 *P. falciparum* clone

RBCs were obtained from 13 Scottish donors (4 donors of group O and A, and 5 for group B) following verbal consent. After the removal of plasma and white cells as described above, RBCs from the 13 donors, and an aliquot of the uninfected group O RBCs in which the parasites were cultured (hereafter known as 'home O') were labelled with 6-carboxyfluorescein diacetate (C-FDA) (150μg/ml in RPMI 1640 medium), at 2% haematocrit for 15 minutes, washed twice with RPMI 1640 and stored at 4˚C. Blood group preference assays were carried out as described [16,17,37]. Briefly, IT/R29 parasites in RPMI 1640 at 6–10% parasitaemia and a rosette frequency of >60% were stained with 25μg/ml ethidium bromide for 5 minutes with 200μg/ml fucoidan added to disrupt all rosettes [75]. Labelled uninfected RBCs were also resuspended at 2% haematocrit in RPMI 1640 with 200μg/ml fucoidan. Equal volumes of parasite culture and labelled cells were mixed in triplicate, and the percentage labelled cells in the resulting mixtures counted (300 cells). After 3 washes, cells were resuspended in RPMI 1640 with 10% AB serum and incubated for 1 hour at 37˚C to allow rosettes to reform. For each replicate two counts were made: (1) the percentage labelled cells in the mix (300 cells, using only those cells not in rosettes) and (2) the percentage labelled cells in rosettes (200 cells, only those within rosettes). Graphs show the difference between the percentage labelled cells in rosettes and in the mix (using the mean of the 2 counts for % labelled cells in the mix, before and after washes). The mean and SEM of the triplicate readings are shown for each RBC donor. For statistical analysis of the differences between blood groups, the triplicate values for each donor were averaged and treated as a single data point, such that n = 4 for groups O and A, and n = 5 for group B. Blood groups were compared using a Kruskal Wallis test with Dunn's multiple comparisons using Prism v7.0 (Graphpad Inc., San Diego, California).

## Results

### ABO genotypes were strongly concordant with ABO phenotypes in Kenyan children

*ABO* genotype was determined in a case-control study on susceptibility to severe malaria involving >5000 Kenyan children, while ABO blood group phenotypes were assessed on a sub-group of 2761 control children. O was the most common blood group being found in 55.8% of controls, followed by A (23.1%) and B (18.2%), with AB being relatively rare (2.9%) (Table 1). In terms of genotype, *OO* was the most common (54.8% of controls), followed by *AO* (21.0%) and *BO* (17.7%). The *AA* (2.1%), *BB* (1.4%) and *AB* (3.0%) genotypes were rare (Table 1). The overall agreement between genotype and phenotype was 97.1% (Kappa score, 0.95; *p* <0.0001), giving confidence that studies using *ABO* genotype to infer ABO blood group phenotype are based on solid assumptions. The highest agreement was seen for the *OO*

**Table 2.** *ABO* genotype-phenotype agreement[§].

| ABO phenotype | ABO genotype | | | | | |
|---|---|---|---|---|---|---|
| | *OO*<br>N (%) | *AO*<br>N (%) | *AA*<br>N (%) | *BO*<br>N (%) | *BB*<br>N (%) | *AB*<br>N (%) |
| O | **1453 (99.3)** | 25 (4.3) | 1 (2.0) | 13 (2.8) | 3 (7.3) | 0 (0.0) |
| A | 4 (0.3) | **549 (94.6)** | **54 (98.0)** | 17 (3.6) | 0 (0.0) | 1 (1.4) |
| B | 5 (0.3) | 5 (0.9) | 0 (0.0) | **442 (93.4)** | **38 (92.7)** | 2 (2.7) |
| AB | 1 (0.1) | 1 (0.2) | 0 (0.0) | 1 (0.2) | 0 (0.0) | **71 (95.9)** |

[§]Agreement between ABO blood group phenotype and *ABO* genotype was determined for 2686 community control samples that were successfully typed by both methods. The expected genotype-phenotype combinations are highlighted in bold. Overall concordance using Cohen's Kappa statistic was 0.95, $p<0.0001$ (a score of 1 corresponds to complete agreement, while 0 corresponds to no agreement).

genotype, with 99.3% of *OO* samples being phenotyped as blood group O (Table 2). The blood group A- and AB-encoding genotypes (*AO*, *AA*, *AB*) showed >94% genotype-phenotype concordance. The blood group B encoding genotypes (*BO* and *BB*) showed the lowest concordance at 93.4% and 92.7% respectively, with most of the discordant samples being typed as blood group O or A by serology (Table 2).

## The *BB* and *AB* genotypes were associated with the highest odds ratios for severe malaria in the case-control study

We next examined the relationship between *ABO* genotype and susceptibility to severe malaria in the case-control study. Associations between ABO blood group and severe malaria for this full dataset, which also included cases who we have subsequently shown to have a low probability of severe malaria, have been reported previously [14]. However, our current analysis differs by using a more precise case-definition of severe malaria [47], and by focussing on differences between specific non-*O* genotypes, which were not analysed in the previous study [14]. The general demographic characteristics of the study participants are presented in Table 3. Cases and controls differed in ethnic composition; hence, this variable was included in subsequent analyses. Cases and controls also differed in age, which is a consequence of

**Table 3.** **General characteristics for the Kenyan case-control study.**

| | Controls | Cases | *p* value |
|---|---|---|---|
| Kenya | n = 3949 | n = 1403 (all severe malaria) | |
| Gender | | | |
| Males | 1992 (50.4%) | 717 (51.1%) | |
| Females | 1957 (49.6%) | 686 (48.9%) | 0.670 |
| Ethnicity | | | |
| Giriama | 1836 (46.5%) | 842 (60.0%) | |
| Chonyi | 1408 (35.7%) | 317 (22.6%) | |
| Kauma | 450 (11.4%) | 94 (6.7%) | |
| Others | 255 (6.5%) | 150 (10.7%) | **<0.001** |
| Age in months*<br>Median (IQR) | 6 (5–8) | 29 (17–44) | **<0.001** |

Comparisons between cases and controls were performed using the Pearson's Chi square test for gender and ethnic group while the Kruskal Wallis test was used to test for differences in age (as a continuous variable). IQR, interquartile range.

study design, as described in the methods section. *ABO* genotype distributions in relation to the variables included in this study are shown in S1 Table. A significantly higher proportion of the controls had the *OO* genotype compared to the severe malaria cases (S1 Table). There were minor differences in *ABO* genotype distribution in relation to gender in controls, which was unexpected and may be a chance finding (S1 Table). Parasite density in the severe malaria cases did not vary significantly across the different *ABO* genotypes (S2 Table).

We tested for associations between *ABO* genotype and severe malaria, including the specific severe malaria syndromes cerebral malaria (CM), severe malarial anaemia (SMA) and respiratory distress (RD), and malaria-specific mortality using a logistic regression model both with, and without, adjustment for the confounders HbS, $\alpha^+$thalassaemia, gender, and ethnicity. When grouped together, the non-*O* genotypes were associated with a 49% greater risk of all severe malaria (adjusted ORs (aOR) 1.49; 95% CI 1.31–1.70; $p<0.001$), with similar values for each of the clinical sub-phenotypes individually (Table 4). When considered separately, the *BB* and *AB* genotypes were associated with the highest risks of severe malaria, 108% and 93% greater risk respectively, compared to *OO* genotype (*BB* aOR 2.08; 95% CI 1.29–3.37; $p = 0.003$ and *AB* aOR 1.93; 95% CI 1.37–2.72; $p<0.001$), and were also associated with the highest risks for the specific severe malaria syndromes of CM and RD (*BB*) and SMA (*AB*) (Table 4).

To examine the hypothesis that RBC A/B antigen levels affect malaria susceptibility, we used the Wald test to analyse differences in ORs between single (*AO*, *BO*) and double dose (*AA*, *AB*, *BB*) non-*O* genotypes generated by logistic regression analysis above using the *OO* genotype as the reference group. When compared to *OO* genotype, *AA* and *AB* were associated with a 46% (aOR 1.46; 95% CI 1.29–3.37; $p = 0.003$) and 93% greater risk of severe malaria overall respectively compared to 27% greater risk seen with *AO*. Similarly, *BB* (aOR 2.08) and *AB* (aOR 1.93) were associated with higher odds for all severe malaria that that seen with *BO* (aOR *1.65)*. However, only the *AB* vs *AO* comparison reached statistical significance ($p = 0.020$, Table 5). Similarly, for the specific severe malaria syndromes, double dose *A* and/or *B* genotypes were associated with higher ORs than single dose genotypes in many cases but were not statistically significant (S3 Table). As an alternative approach to testing our hypothesis, we carried out logistic regression analysis to directly compare the ORs of severe malaria for double dose genotypes *AA* and *AB* blood groups to single dose *AO* as reference and similarly compared *BB* and *AB* to *BO* as reference. Mirroring the Wald test results, double dose non-*O* genotypes were associated with higher ORs when compared to single dose *AO* or *BO* genotypes, but only the *AB* vs *AO* comparison was statistically significant (S4 Table). Overall, the data show patterns that are consistent with the hypothesis, but do not allow us to reject the null hypothesis.

### *AA/AB* RBCs form larger rosettes than *OO* RBCs with a blood group A-preferring *P. falciparum* clone

To determine whether host RBC *ABO* genotype influences *P. falciparum* rosetting, we examined rosette size following parasite invasion into RBCs from 60 donors (*OO* = 23, *AO* = 18, *BO* = 9, *AA* = 2, *BB* = 1, *AB* = 7) using the *P. falciparum* clone IT/R29 [64]. Preliminary experiments confirmed a strong blood group A-preference in this parasite line, with a lesser preference for group B RBCs (S1 Fig) [19,26]. Following parasite invasion and maturation in RBCs from the 60 *ABO* genotyped donors, the double dose genotype RBCs (*AA*, *BB*, *AB)* formed significantly larger rosettes and showed a higher proportion of large rosettes compared to single dose (*AO*, *BO*) and *OO* genotype RBCs (Fig 1). In contrast, the single dose *AO* and *BO* genotype RBCs did not differ from *OO* genotype RBCs in terms of rosette size or the proportion of

**Table 4. Case-control analysis of the association between *ABO* genotype and severe malaria syndromes.**

| Case Phenotype | No. Cases/controls | *ABO* genotype | Crude | | | | Adjusted[†] | | | |
|---|---|---|---|---|---|---|---|---|---|---|
| | | | OR | LCI | UCI | *p* value | OR | LCI | UCI | *p* value |
| *All SM* | 623/2126 | *OO* | 1 | | | | 1 | | | Reference |
| | 306/810 | *AO* | 1.29 | 1.10 | 1.51 | 0.002 | 1.27 | 1.07 | 1.50 | 0.006 |
| | 37/83 | *AA* | 1.52 | 1.02 | 2.26 | 0.039 | 1.46 | 0.95 | 2.22 | 0.082 |
| | 61/115 | *AB* | 1.81 | 1.31 | 2.50 | < 0.001 | 1.93 | 1.37 | 2.72 | <0.001 |
| | 337/683 | *BO* | 1.68 | 1.44 | 1.97 | < 0.001 | 1.65 | 1.40 | 1.95 | <0.001 |
| | 34/55 | *BB* | 2.11 | 1.36 | 3.27 | 0.001 | 2.08 | 1.29 | 3.37 | 0.003 |
| | 775/1746 | Non-*O* | 1.52 | 1.34 | 1.71 | < 0.001 | 1.49 | 1.31 | 1.70 | <0.001 |
| *All CM* | 326/2126 | *OO* | 1 | | | | 1 | | | Reference |
| | 166/810 | *AO* | 1.34 | 1.09 | 1.64 | 0.005 | 1.29 | 1.05 | 1.60 | 0.018 |
| | 22/83 | *AA* | 1.73 | 1.07 | 2.81 | 0.027 | 1.60 | 0.96 | 2.67 | 0.070 |
| | 26/115 | *AB* | 1.47 | 0.95 | 2.29 | 0.085 | 1.47 | 0.93 | 2.34 | 0.101 |
| | 160/683 | *BO* | 1.53 | 1.24 | 1.88 | < 0.001 | 1.45 | 1.17 | 1.80 | 0.001 |
| | 16/55 | *BB* | 1.90 | 1.07 | 3.35 | 0.027 | 1.90 | 1.03 | 3.49 | 0.039 |
| | 390/1746 | Non-*O* | 1.46 | 1.24 | 1.71 | < 0.001 | 1.40 | 1.18 | 1.65 | <0.001 |
| *All SMA* | 181/2126 | *OO* | 1 | | | | 1 | | | Reference |
| | 97/810 | *AO* | 1.41 | 1.09 | 1.82 | 0.010 | 1.35 | 1.03 | 1.78 | 0.031 |
| | 9/83 | *AA* | 1.27 | 0.63 | 2.58 | 0.773 | 1.18 | 0.56 | 2.52 | 0.662 |
| | 19/115 | *AB* | 1.94 | 1.17 | 3.32 | 0.011 | 2.05 | 1.21 | 3.45 | 0.007 |
| | 99/683 | *BO* | 1.70 | 1.31 | 2.21 | < 0.001 | 1.62 | 1.23 | 2.12 | 0.001 |
| | 10/55 | *BB* | 2.14 | 1.07 | 4.26 | 0.031 | 1.71 | 0.75 | 3.89 | 0.204 |
| | 234/1746 | Non-*O* | 1.57 | 1.28 | 1.93 | < 0.001 | 1.50 | 1.21 | 1.86 | <0.001 |
| *All RD** | 181/2126 | *OO* | 1 | | | | 1 | | | Reference |
| | 101/810 | *AO* | 1.47 | 1.13 | 1.89 | 0.004 | 1.41 | 1.08 | 1.83 | 0.012 |
| | 13/83 | *AA* | 1.84 | 1.01 | 3.37 | 0.048 | 1.68 | 0.89 | 3.17 | 0.113 |
| | 12/115 | *AB* | 1.23 | 0.66 | 2.26 | 0.516 | 1.25 | 0.67 | 2.33 | 0.489 |
| | 99/683 | *BO* | 1.70 | 1.31 | 2.21 | < 0.001 | 1.63 | 1.25 | 2.13 | <0.001 |
| | 11/55 | *BB* | 2.35 | 1.21 | 4.57 | 0.012 | 2.01 | 0.96 | 4.21 | 0.064 |
| | 236/1746 | Non-*O* | 1.59 | 1.30 | 1.95 | < 0.001 | 1.51 | 1.23 | 1.87 | <0.001 |
| *Mortality** | 52/2126 | *OO* | 1 | | | | 1 | | | Reference |
| | 33/810 | *AO* | 1.67 | 1.07 | 2.60 | 0.024 | 1.74 | 1.11 | 2.73 | 0.017 |
| | 2/83 | *AA* | 0.99 | 0.24 | 4.11 | 0.984 | 1.00 | 0.24 | 4.21 | 0.999 |
| | 4/115 | *AB* | 1.42 | 0.51 | 4.00 | 0.505 | 1.50 | 0.53 | 4.26 | 0.450 |
| | 23/683 | *BO* | 1.38 | 0.84 | 2.27 | 0.209 | 1.35 | 0.81 | 2.25 | 0.258 |
| | 2/55 | *BB* | 1.49 | 0.35 | 6.26 | 0.589 | 1.66 | 0.39 | 7.08 | 0.497 |
| | 64/1746 | Non-*O* | 1.50 | 1.03 | 2.17 | 0.033 | 1.53 | 1.04 | 2.24 | 0.029 |

SM: Severe malaria; CM: cerebral malaria; SMA: severe malaria anaemia; RD: respiratory distress; OR: Odds Ratio; LCI: Lower Confidence Interval (95%); UCI: Upper Confidence Interval; P: P-value using a logistic regression model.

[†]adjusted for HbS, α+thalassaemia, gender, ethnicity and interaction (HbS and α+thalassaemia).

* no interaction term included between HbS and α+thalassaemia genotype.

large rosettes (Fig 1). Multivariate regression analysis examining each individual *ABO* genotype and adjusting for confounding variables showed that only the double dose *AA* and *AB* genotype RBCs differed significantly from *OO* genotype RBCs in terms of mean rosette size and proportion of large rosettes (Table 6 and S2 Fig).

**Table 5. A comparison of the odds ratio differences for severe malaria between single dose and double dose non-*O* genotypes using the Wald test.**

| Case Phenotype | *ABO* genotype | No. Cases/Controls | Odds Ratio comparisons | Wald test *p* value |
|---|---|---|---|---|
| *All SM* | *AO vs AA* | 306/810; 37/83 | 1.27/1.46 | 0.528 |
| | *BO vs BB* | 337/683; 34/55 | 1.65/2.08 | 0.351 |
| | *AO vs AB* | 306/810; 61/115 | 1.27/1.93 | 0.020 |
| | *BO vs AB* | 337/683; 61/115 | 1.65/1.93 | 0.384 |

Single dose and double dose non-*O* genotype odds ratios for severe malaria used in the Wald test comparisons were generated following a fixed-effects logistic regression model comparing genotype frequencies between the non-*O* genotypes to the reference *OO* genotype with adjustments for self-reported ethnicity, gender, $\alpha^+$thalassaemia and HbAS.

## *P. falciparum* cytoadhesion and PfEMP1 display do not differ between ABO genotypes

Effects on other parasite adhesion-related properties, such as reduced cytoadhesion to microvascular endothelial cells via ICAM-1 and CD36 receptors and reduced display of PfEMP1 on the surface of iRBCs, have been implicated as mechanisms of protection for various human RBC polymorphisms [41,76]. To determine whether *ABO* genotype might influence malaria

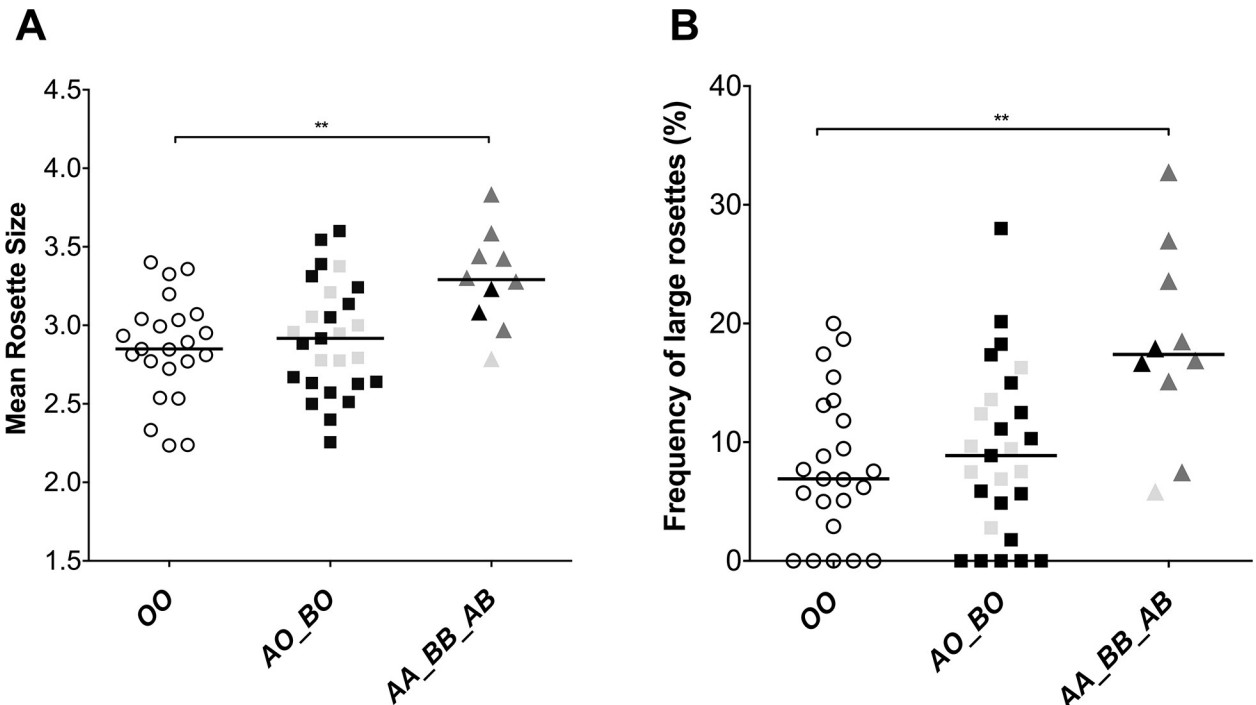

**Fig 1. *P. falciparum* IT/R29 rosette size and frequency of large rosettes by *ABO* genotype. (A)** IT/R29 mean rosette size (number of uninfected RBCs per rosette) **(B)** IT/R29 frequency of large rosettes (more than 4 uninfected RBCs per rosette). Purified IT/R29 infected RBCs (iRBCs) were allowed to invade into RBCs from 60 donors (*OO* n = 23 white circles, *AO* n = 18 black squares, *BO* n = 9 grey squares, *AA* n = 2 black triangles, *BB* n = 1 grey triangles, *AB* n = 7 dark grey triangles) and rosette size and frequency of large rosettes were assessed the next day by fluorescence microscopy. Samples were tested over two consecutive experimental days (day 1 = 30 and day 2 = 30) in duplicates. Horizontal bars represent the median rosette size and median frequency of large rosettes single dose (*AO*, *BO*) and double dose (*AA*, *BB*, *AB*) genotypes. The number of donors per genotype are shown in parenthesis. Sample genotype was masked during counting to avoid observer bias. ** Double dose non-*O* genotypes differed from *OO* in mean rosette size (Kruskal Wallis test with Dunn's multiple comparisons p = 0.0065) and frequency of large rosettes (Kruskal Wallis test with Dunn's multiple comparisons p = 0.0068).

**Table 6. Rosetting of *P. falciparum* clone IT/R29 by *ABO* genotype.**

| IT/R29 rosette size | | | | |
|---|---|---|---|---|
| **N** | **Genotype** | **Mean rosette size** | **95% CI** | ***p* value** |
| 23 | *OO* | 2.89 | 2.77–3.01 | - |
| 18 | *AO* | 2.92 | 2.78–3.05 | 0.737 |
| 2 | *AA* | 3.36 | 2.95–3.78 | 0.030 |
| 9 | *BO* | 2.92 | 2.73–3.11 | 0.773 |
| 1 | *BB* | 2.52 | 1.93–3.11 | 0.229 |
| 7 | *AB* | 3.28 | 3.06–3.50 | 0.003 |
| 37 | *Non-O* | 3.00 | 2.92–3.11 | 0.157 |
| **Frequency of IT/R29 large rosettes** | | | | |
| **N** | **Genotype** | **% of large rosettes (>4 uninfected RBCs/rosette)** | **95% CI** | ***p* value** |
| 23 | *OO* | 8.49 | 5.75–1.22 | - |
| 18 | *AO* | 9.36 | 6.27–12.22 | 0.674 |
| 2 | *AA* | 20.55 | 11.20–29.91 | 0.016 |
| 9 | *BO* | 8.45 | 4.09–12.81 | 0.988 |
| 1 | *BB* | 1.29 | -12.01–14.59 | 0.296 |
| 7 | *AB* | 18.23 | 13.19–23.28 | 0.001 |
| 37 | *Non-O* | 11.27 | 8.89–13.66 | 0.143 |

Differences in IT/R29 rosette size or frequency of large rosettes by *ABO* genotype were tested using multivariate regression analysis with adjustment for confounding by HbAS. 60 RBC donor samples were tested once in duplicate over two successive experimental days (day 1 n = 30 and day 2 n = 30), therefore, experimental day was included as a co-variate to account for day-to-day variation.

susceptibility via these mechanisms, we conducted *in vitro* experiments examining cytoadhesion to ICAM-1 and CD36 and PfEMP1 display in relation to infected RBC *ABO* genotype. No genotype-specific differences were seen (S3 and S4 Figs and S5 and S6 Tables).

## ABO genotype is not associated with risk of either uncomplicated malaria or asymptomatic infection

Rosetting is primarily a property of parasite isolates causing severe malaria, and no (or very low level) rosetting is seen in *P. falciparum* isolates collected from patients with uncomplicated malaria [28,77]. If *ABO* genotype influences malaria susceptibility via a mechanism related to rosette size and microvascular obstruction, we would predict that *ABO* genotype associations would only be demonstrated in severe disease, and not in mild or asymptomatic infections. We tested this hypothesis using a longitudinal cohort study and cross-sectional surveys carried out in the same geographic area as the case-control study. No significant associations were seen between *ABO* genotype and either uncomplicated malaria or asymptomatic *P. falciparum* infection (S7–S9 Tables).

## Discussion

Although associations between ABO blood group, *P. falciparum* rosetting and susceptibility to severe malaria are well-established [9–11,16,20,28,78–80], to the best of our knowledge, higher resolution analyses to determine their associations with distinct *ABO* genotypes have not been conducted. Our results show a significant effect of *ABO* genotype on rosette size, with double dose *AA/AB* genotype RBCs forming significantly larger rosettes than *OO* genotype RBCs with the A-preferring *P. falciparum* line IT/R29. In contrast, single dose *AO* and *BO* genotype RBCs

did not differ from *OO* genotype RBCs in the rosetting assays. Previous work has shown that rosettes in non-O blood are not only larger but are also more stable and more difficult to disrupt with reagents such as heparin, compared to rosettes in group O RBCs, [16,18,19,26,29]. Rosette size and stability are of great potential importance in malaria pathology, because experimental studies show that larger rosettes in A compared to O RBCs are more resistant to disruption under shear stress and therefore cause greater obstruction to capillary-sized channels [27]. Hence, a direct causal link between host *ABO* genotype, parasite virulence in terms of rosette size, and the primary pathological process of microvascular obstruction in severe malaria is suggested.

Our finding that the *ABO* gene dose affects rosetting supports our prediction that *AA/BB/AB* individuals will be at higher risk of severe malaria than *AO/BO* individuals. In the case-control study we found that for most comparisons, *AA/BB/AB* individuals had higher ORs for severe malaria and specific severe malaria syndromes than *AO* and *BO* individuals, but we could not reject the null hypothesis on statistical grounds. Despite recruiting ~1400 severe malaria cases into our study, the frequency of the key *AA*, *BB* and *AB* genotypes in the dataset was low (37, 34 and 61 severe cases respectively) which limited the power of our study. Similarly, the *in vitro* rosetting assays, which used RBCs collected from the same study area as the epidemiological work, were limited by small sample size in the double dose genotypes (*AA*, *AB* and *BB)*. Future studies with larger sample sizes in the key double dose genotypes will be needed to confirm our findings.

Our *in vitro* rosetting results support previous suggestions that rosetting is a causal factor in the protective association between ABO blood group and severe malaria [7,16,25,29]. However, alternative protective mechanisms have been suggested for other polymorphisms [41,81]. We examined two of these mechanisms here, iRBC binding to endothelial receptors (ICAM-1 and CD36) and PfEMP1 expression levels on the surface of iRBCs. We found no evidence for an association between *ABO* genotype and either endothelial receptor binding or PfEMP1 display. However, our study only used a single parasite clone (ItG, expressing the ITvar16 PfEMP1 variant) and two endothelial receptors. Future studies could further explore the role of ABO on cytoadhesion using other *P. falciparum* clones/variants and additional endothelial receptors. The only previous study to examine ABO and cytoadhesion found no effect of ABO serological phenotype on binding to multiple endothelial receptors, with decreased ICAM-1 adhesion in non-O blood groups being the only significant result [82], a finding that is inconsistent with the association between non-O and increased risk of severe malaria. Other protective mechanisms for ABO have also been suggested, including effects on RBC invasion [83,84] and phagocytic clearance of iRBCs [26,85,86]. However, both invasion- and clearance-related mechanisms would be expected to have an impact by lowering parasite burden, yet no consistent effect of ABO blood group on parasite density has been seen across multiple studies (reviewed by [80]), and we found no significant association in our case-control study (S2 Table). Taken together, existing data support a role for reduced *P. falciparum* rosetting in O RBCs and consequent reduced microvascular obstruction, as a key mechanism by which ABO blood group influences the risk of severe malaria.

An additional aim of our study was to investigate the correlation between *ABO* genotype and blood group phenotype in an African population. We report strong agreement between *ABO* genotype and phenotype, with only 79/2686 samples (2.9%) being discordant. The strongest concordance was for genotype *OO* with blood group O (99.3%), and the weakest for genotype *BB* with blood group B (92.3%). Two American studies using the same SNPs reported a genotype-phenotype concordance of 100% and 92% respectively, although the perfect concordance seen by Risch and colleagues could potentially have been a reflection of their very small sample size (n = 30) [33,34]. Some discrepancies in our study, for example, the 42 samples

genotyped as *AO*, *AA*, *BO* and *BB* but serologically typed as blood group O, could be due to weak A and B subgroups that reduce A and B antigen density on the RBC surface [87–89], which might have resulted in samples being typed as O in standard agglutination assays [90–93]. Another possible source of error might have resulted from the assumption made in inferring *ABO* genotype in heterozygotes. 62/79 (78%) of discrepant samples were heterozygous for the *O* deletional allele ("GD" in Table 1). The double heterozygote combination of GD at rs8176719 and AC at rs8176746 can result in either *BO* genotype (GA and DC haplotype) or *AO* genotype from the rare GC and DA haplotype. The *O* deletion is thought to have arisen on the background of an *A* allele [2], so all individuals were assigned as *BO* genotype. However, *O* deletions arising on a *B* allele background have been described [94], and could account for some of the discrepancies we observed. Finally, it is also possible that some samples were erroneously typed by serology. We were unable to access fresh RBC samples from the same individuals to repeat the typing.

## Conclusions

In conclusion, our combined epidemiological and laboratory studies support the hypothesis that *AO/BO* heterozygotes differ from *AA/AB/BB* individuals in relation to *P. falciparum* rosetting and severe malaria risk. Alternative mechanisms of protection for blood group O and the *OO* genotype, such as binding to endothelial receptors ICAM-1 and CD36 and effects on PfEMP1 display were not supported by the data. Additional studies examining the effects of *ABO* genotype, as well as weak A and B blood groups, may give further insights into the complex host-parasite interactions in severe malaria.

## Supporting information

**S1 Table. The general characteristics of children recruited to the case-control study by *ABO* genotype.**
(PDF)

**S2 Table. *P. falciparum* parasite density by *ABO* genotype in severe malaria cases.**
(PDF)

**S3 Table. Comparing Odds ratio differences for specific severe malaria syndromes between single dose and double dose non-*O* genotypes using the Wald test.**
(PDF)

**S4 Table. A comparison of the odds ratio differences for severe malaria between single dose and double dose non-*O* genotypes using logistic regression with single dose non-*O* genotypes *AO/BO* as reference.**
(PDF)

**S5 Table. Cytoadhesion of *P. falciparum* line ItG by *ABO* genotype.**
(PDF)

**S6 Table. ITvar9 PfEMP1 expression by *ABO* genotype.**
(PDF)

**S7 Table. General characteristics of the Kilifi longitudinal cohort study by *ABO* genotype.**
(PDF)

**S8 Table. Incidence rate ratios (IRR) for uncomplicated malaria in Kenya by *ABO* genotype.**
(PDF)

**S9 Table. Odds ratios (OR) for asymptomatic malaria in Kenya by *ABO* genotype.**
(PDF)

**S1 Fig. *P. falciparum* IT/R29 ABO blood group *rosetting* preference.** ABO blood group preference assay of the IT/R29 parasite clone was assessed using fluorescently-labelled RBC from 13 donors (4 group O, 4 group A and 5 group B). "Home O" indicates the blood group O donor that was used to culture the parasites. The y-axis shows the difference between the percentage of labelled cells found in rosettes, compared to the percentage found in the mix. Experiments were carried out in triplicate, and the mean and SEM are shown for each donor. For statistical analysis, the triplicate values for each donor were averaged and treated as a single data point, such that n = 4 for groups O and A, and n = 5 for group B. The blood groups were compared using a Kruskal Wallis test with Dunn's multiple comparisons (** $p < 0.01$).
(TIFF)

**S2 Fig. *P. falciparum* IT/R29 rosette size and frequency of large rosettes by *ABO* genotype.**
**(A)** IT/R29 mean rosette size (number of uninfected RBCs per rosette) **(B)** IT/R29 frequency of large rosettes (more than 4 uninfected RBCs per rosette). Purified IT/R29 infected RBCs (iRBCs) were allowed to invade into RBCs from 60 donors (*OO* n = 23, *AO* n = 18, *BO* n = 9, *AA* n = 2, *BB* n = 1, *AB* n = 7) and rosette size and frequency of large rosettes were assessed after one complete life-cycle by fluorescence microscopy. Samples were tested over two consecutive experimental days (day 1 = 30 and day 2 = 30) in duplicates. Horizontal bars represent the median rosette size and median frequency of large rosettes for each genotype. The number of donors per genotype are shown in parenthesis. Sample genotype was masked during counting to avoid observer bias. **AB differed from *OO* for both mean rosette size and frequency of large rosettes, $p < 0.01$ Kruskal-Wallis test with Dunn's multiple comparisons.
(TIFF)

**S3 Fig. Relative cytoadherence of *P. falciparum* ItG by *ABO* genotype. (A)** Relative binding to CD36 recombinant protein. **(B)** Relative binding to ICAM-1 recombinant protein. Purified ITg infected RBCs (iRBCs) were allowed to invade into RBCs from 112 donors (*OO* n = 51, *AO* n = 32, *BO* n = 17, *AA* n = 6, *BB* n = 3, *AB* n = 3) and static adhesion to immobilized recombinant proteins was tested after one complete life-cycle. Samples were tested over seven experimental days (day 1 n = 13, day 2 n = 8, day 3 n = 12, day 4 n = 1, day 5 n = 55, day 6 n = 15 and day 7 n = 8), with each donor being tested once. For each RBC sample, adhesion was tested in two dishes with triplicate protein spots in each dish and the data presented as the mean iRBC bound/mm$^2$ for each donor. Because baseline binding using a single donor varies from day to day, the binding data for each sample were normalized to that of the mean binding for the control iRBCs (*OO*) run on the same day (number of reference *OO* genotype samples run each day; day 1 n = 7, day 2 n = 4, day 3 n = 8, day 4 n = 1, day 5 n = 23, day 6 n = 7 and day 7 n = 2). Horizontal bars represent median relative adhesion for each genotype. Number of samples per genotype are shown in parenthesis. A Kruskal-Wallis test gave $p = 0.310$ for CD36 adhesion and $p = 0.814$ for ICAM-1 adhesion.
(TIFF)

**S4 Fig. Relative PfEMP1 expression in *P. falciparum* IT/R29 by *ABO* genotype. (A)** Relative ITvar9 PfEMP1 expression. **(B)** Relative ITvar9 positive iRBCs. Purified IT/R29 infected RBCs (iRBCs) were allowed to invade into RBCs from 60 donors (*OO* n = 23, *AO* n = 18, *BO* n = 9, *AA* n = 2, *BB* n = 1, *AB* n = 7) and PfEMP1 expression was assessed after one complete life-cycle by flow cytometry with antibodies specific for the ITvar9 PfEMP1 variant. Samples were tested over two consecutive experimental days (day 1 = 30 and day 2 = 30). Median fluorescent

intensity and proportion of ITvar9 positive iRBC data for all samples were normalized to that of the mean MFI and proportion of ITvar9 positive iRBCs for the control pRBCs (*OO*) (14 and 9 reference *OO* genotype samples run on day 1 and 2 respectively) run on the same day. Horizontal bars represent median relative MFI and proportion of ITvar9 positive iRBCs for each genotype. Kruskal-Wallis test $p = 0.076$ for MFI and $p = 0.285$ for % positive iRBCs. (TIFF)

## Acknowledgments

We are grateful to all the children who donated samples to this study, and their parents and guardians. We are also grateful to the clinical, field and laboratory staff at the KEMRI-Wellcome Trust Research Programme, including the laboratory team for the Human Genetics Group. We also thank Professor Alister Craig (Liverpool School of Tropical Medicine) for his generous contribution of the ICAM-1-Fc purified recombinant protein.

This paper was published with permission from the Director of the Kenya Medical Research Institute (KEMRI).

## Author Contributions

**Conceptualization:** D. Herbert Opi, Thomas N. Williams, J. Alexandra Rowe.

**Data curation:** Carolyne M. Ndila, Lucy B. Ochola, Gideon Nyutu, Bethseba R. Siddondo, Gavin Band, Kirk A. Rockett.

**Formal analysis:** D. Herbert Opi, Carolyne M. Ndila, Thomas N. Williams, J. Alexandra Rowe.

**Funding acquisition:** Dominic P. Kwiatkowski, Thomas N. Williams, J. Alexandra Rowe.

**Investigation:** D. Herbert Opi, Carolyne M. Ndila, Sophie Uyoga, Alex W. Macharia, Clare Fennell, Gideon Nyutu, John Ojal, Mohammed Shebe, Kennedy O. Awuondo, Neema Mturi, Norbert Peshu, Benjamin Tsofa, Gavin Band, Kathryn Maitland, Dominic P. Kwiatkowski, Kirk A. Rockett, Thomas N. Williams, J. Alexandra Rowe.

**Methodology:** D. Herbert Opi.

**Supervision:** Thomas N. Williams, J. Alexandra Rowe.

**Validation:** Thomas N. Williams, J. Alexandra Rowe.

**Writing – original draft:** D. Herbert Opi, Thomas N. Williams, J. Alexandra Rowe.

**Writing – review & editing:** D. Herbert Opi, Carolyne M. Ndila, Sophie Uyoga, Alex W. Macharia, Clare Fennell, Gideon Nyutu, John Ojal, Mohammed Shebe, Kennedy O. Awuondo, Neema Mturi, Norbert Peshu, Benjamin Tsofa, Gavin Band, Kathryn Maitland, Dominic P. Kwiatkowski, Kirk A. Rockett, Thomas N. Williams, J. Alexandra Rowe.

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
