## [Decision Letter · Decision Letter 0]

23 Feb 2023

Dear Dr Herbert Opi, 

Thank you very much for submitting your Research Article entitled 'Non-O ABO blood group genotypes differ in their associations with Plasmodium falciparum rosetting and severe malaria' to PLOS Genetics.

The manuscript was fully evaluated at the editorial level and by independent peer reviewers. The reviewers appreciated the attention to an important problem, but raised some substantial concerns about the current manuscript. Based on the reviews, we will not be able to accept this version of the manuscript, but we would be willing to review a much-revised version. We cannot, of course, promise publication at that time.

If you decide to revise the manuscript for further consideration at PLOS Genetics, please aim to resubmit within the next 60 days, unless it will take extra time to address the concerns of the reviewers, in which case we would appreciate an expected resubmission date by email to plosgenetics@plos.org.

We are sorry that we cannot be more positive about your manuscript at this stage. Please do not hesitate to contact us if you have any concerns or questions.

Yours sincerely,

Susana Campino, PhD

Academic Editor

PLOS Genetics

Hua Tang

Section Editor

PLOS Genetics

Reviewer's Responses to Questions

**Comments to the Authors:**

Reviewer #1: Non-ABO blood group genotypes differ in their associations with Plasmodium falciparum rosetting and severe malaria

This work described tackles a key question in malaria pathogenesis, does ABO blood type influence clinical features of disease? The work is focused partially around determining an association between Non-O dosage and clinical outcome/risk of severe disease and partially around investigating an association between ABO blood type and rosetting, a feature of malaria infection where parasitized RBCs will “coat” themselves in uninfected RBCs.

Overall, this work is interesting and exciting to the field. The results are conveyed with clarity, and the caveats regarding the work well highlighted. Of note is the association study which ultimately stops short of demonstrating an association between O-dosage and the outcome of infection, though replicates a link between O vs non-O, and the link between rosette size and non-O dosage.

The major unique factor of the work, to my eye, is the focus on non-O dosage. The advance described is arguably subtle, showing a tendancy for non-O dosage to increase adverse outcome in the association study. As noted, a lack of sample size and numbers in some ABO genotype groups reduced the power of the study.

As I understand the work, there is a link between O dosage and either rosetting or clinical outcome, beyond which has already been reported (i.e. O vs the rest). This is the result of hard won and clearly described data and analysis and is supportive of a role whereby (1) non-O genotype dosage influences the clinical outcome of malaria disease and (2) rosetting phenotypes vary between ABO genotypes. While this appears to be a subtle advance, the careful work described here is robust and should be a springboard for larger studies with better power.

Given the small numbers in Figure 1, and the focus of the paper around non-O dosage it was somewhat surprising to not see the grouping by number of non-O alleles.

The derivation of the IT clones for growth in A+ cells could have been given greater discussion. The IT/R29 clone, it appears that two clones are used in the study (R29 expressing ITvar9 PfEMP1 and another (ItG), clone expressing ITvar16). How may the way these lines were derived impact the findings of the study? These lines, as I understand, were derived through selection under specific conditions. How stable are the phenotypes in each of these clones, and was the phenotype of each line confirmed using controls?

A minor aside; Kappa score is described as 100% to 0% in the methods, and 0.95 listed in the results. I assume that either the score should be described as a proportion of 0 to 1 or the results should be 95%

Reviewer #2: Manuscript Number: PGENETICS-D-22-01112

Full Title: Non-O ABO blood group genotypes differ in their associations with Plasmodium

falciparum rosetting and severe malaria

Summary: This observational study of Kenyan children from a hospital-based case-control study and a community-based cohort examines the associations between genetically-determined ABO genotypes and risk of severe malaria. These associations are supported by in vitro assays of rosette formation in RBCs of different ABO types infected with laboratory strains of P falciparum. While this study does not necessarily present new biology about malaria pathogenesis, it does improve the precision of our understanding of the relationship between ABO genotypes and rosetting phenomena. While the experimental and epidemiological methods are sound, more attention to the interpretation of multiple testing needs to be provided.

Major comments:

1. Rosettes are proposed to contribute to severe malaria pathogenesis by obstructing microvessels. Presumably, rosettes would be susceptible to removal from the circulation by the spleen. Were there any associations between rosetting genotypes and measures of anemia?

2. It is disconcerting to observe non-random associations between ABO genotypes encoded on chromosome 9 and HBA and HBB genotypes encoded on chromosomes 16 and 11 in the population cohort described in Table S5. Are these spurious associations? If not, what is the reason for these associations? Why the age differences by ABO genotypes?

3. In the case-control study described in Table 3, why is ABO genotype associated with sex and age? Perhaps these associations are driven by cases due to relationships with susceptibility to malaria? If so, perhaps the cases and controls need to be analyzed separately, either way an explanation for these unexpected associations needs to be offered.

4. Some acknowledgement, discussion, or statistical adjustment should be made regarding multiple tests performed and the interpretation of p values.

Minor comments:

5. While the statement in line 352 may be true, “To the best of our knowledge, whether host RBC ABO genotype influences P. falciparum rosetting has not been investigated previously.” It is really a stretch especially when you show such a high concordance between ABO genotype and phenotype.

Typos, etc:

6. The subheading in line 298 could be stated in the past tense.

Reviewer #3: This study aimed to analyze the association between ABO blood group genotype and severe malaria in Sub-Saharan Africa. Expanding on previous knowledge that individuals with blood type O are less vulnerable to severe malaria than non-O blood types, the authors hypothesized that “double-dose” genotypes (AA, BB, or AB) would be more vulnerable to severe malaria compared to “single-dose” genotypes (AO, BO, or OO). This is an interesting and important hypothesis that could possibly improve the tailoring of severe malaria treatment and prevention. The manuscript is well-written and straight-forward to follow. The most valuable contribution is the data analysis from a large case-control study in Kenya, finding that children with double-dose genotypes versus single-dose genotypes had higher odds of severe malaria over children with the OO genotype. Importantly, this was also true for separate severe malaria syndromes. However, it will be crucial to see if these results remain robust after age-adjustment. It was also important to learn that ABO genotype may not always concur with phenotype, which should be of interest to the broader field of genetics and genomics.

The authors included additional epidemiological and mechanistic analyses with a more limited contribution to the overall findings. These analyses had small sample sizes and possible statistical overfitting, raising concerns that this in vitro evidence was not substantial enough to support the article’s conclusions.

Major comments

1. The authors’ hypothesis is that double-dose genotypes (AA, BB) are more vulnerable to severe malaria than single-dose genotypes (AO, BO). The authors use Wald tests to determine that the double-dose genotypes had higher odds of severe malaria over OO compared to the single-dose genotypes, but they do not find statistical significance. Why not compare the odds of severe malaria between double-dose and single-dose genotypes directly?

2. In the case-control study, are there age differences between the cases and controls? I would like to see the demographic characteristics stratified by cases/controls. It seems that age is not controlled for and, given that there are age differences between the genotypes, it may be an important confounding factor to consider. Ref 14 suggests that there may be a significant age difference between cases and controls. The authors did control for age in the cohort and cross-sectional analyses.

1. Were there any episodes of severe malaria in the controls? This information should be stated.

3. In Table 4, the authors used the same controls for each severe malaria phenotype comparison. Again, this makes me wonder whether the control pools are age-comparable to different subsets of cases.

4. In Table 5, the accompanying results text, and the discussion, the authors should be clear that the reference group is still OO in all odds ratios presented. Rather than comparing single-dose to double-dose genotypes directly, they are comparing the magnitude of each genotype to OO.

5. The sample sizes of RBCs for the in vitro rosetting and receptor adhesion assays were very small for the AA, BB, and AB genotypes, reducing the reliability of the statistical findings. The authors should qualify their conclusions about static adhesion and PfEMP1 expression, as their findings are limited to two PfEMP1 variants (particularly Lines 415-416).

1. Consider changing Line 423 to “a key mechanism”, not “the key mechanism”

6. Related to above, the authors adjusted the in vitro signals for several covariates, but I am not sure that the sample size can support so many parameters without overfitting the model.

7. While the ABO findings in the cohort study are interesting, they would be more informative if there was specific information about rosetting in the symptomatic and asymptomatic episodes. As such, the conclusions from this section appear limited and somewhat peripheral to the rest of the study.

8. Are there instances of clinically significant ABO genotype-phenotype discrepancies? As this is a motivating concern underlying this study, any examples should be cited and explained in the Introduction and/or Discussion.

9. The Discussion should comment on the specific findings for particular severe malaria syndromes and ABO genotypes.

Minor comments

• Background

1. It should be mentioned that RIFINs and STEVORs also mediate rosetting, especially RIFINs in blood type A.

2. The cohort and cross-sectional analyses were not introduced.

• Results

1. I don’t understand what the authors were trying to show by demonstrating Hardy-Weinberg Equilibrium among the controls in the case-control study. This should be clarified.

2. Reading the text, I was wondering what an example of ABO genotype-phenotype discordancy may look like. It was made clearer in Table 2 and the discussion, but an example in the text would help.

3. In Table 3, I think that column percentages would support the authors’ argument better than row percentages.

4. The authors should consider providing interpretation of the odds ratios from Table 4. For example, they could indicate that non-O genotypes had 49% higher odds of severe malaria compared to OO. Simply stating that “the odds ratio is 1.49” does not help the reader’s understanding.

5. Since the authors used a parasite strain that preferentially forms rosettes with blood type A, for their rosetting experiments, it seems like the only important result is that AA > AO. Could they derive information about B-antigens if they did not use an A-preferring parasite?

6. The authors should expand on the null findings for the cytoadhesion and cohort study sections in the Discussion, with specific acknowledgment of limitations and next steps.

**Have all data underlying the figures and results presented in the manuscript been provided?**

Reviewer #1: Yes

Reviewer #2: Yes

Reviewer #3: **No: **The datasets from the case-control study and cohort study are not included.

PLOS authors have the option to publish the peer review history of their article (what does this mean?). If published, this will include your full peer review and any attached files.

Reviewer #1: No

Reviewer #2: No

Reviewer #3: No

---

## [Decision Letter · Decision Letter 1]

8 Aug 2023

Dear Dr Herbert Opi, 

We are pleased to inform you that your manuscript entitled "Non-O ABO blood group genotypes differ in their associations with Plasmodium falciparum rosetting and severe malaria" has been editorially accepted for publication in PLOS Genetics. Congratulations!

Yours sincerely,

Susana Campino, PhD

Academic Editor

PLOS Genetics

Hua Tang

Section Editor

PLOS Genetics

Comments from the reviewers (if applicable):

Reviewer's Responses to Questions

**Comments to the Authors:**

Reviewer #2: Thank you for the thorough responses and revisions.

**Have all data underlying the figures and results presented in the manuscript been provided?**

Reviewer #2: Yes

PLOS authors have the option to publish the peer review history of their article (what does this mean?). If published, this will include your full peer review and any attached files.

Reviewer #2: No

**Data Deposition**

http://datadryad.org/submit?journalID=pgenetics&manu=PGENETICS-D-22-01112R1

**Press Queries**

---

## [Editor Report · Acceptance letter]

12 Sep 2023

PGENETICS-D-22-01112R1 

Non-O ABO blood group genotypes differ in their associations with Plasmodium falciparum rosetting and severe malaria 

Dear Dr Opi, 

We are pleased to inform you that your manuscript entitled "Non-O ABO blood group genotypes differ in their associations with Plasmodium falciparum rosetting and severe malaria" has been formally accepted for publication in PLOS Genetics! Your manuscript is now with our production department and you will be notified of the publication date in due course.

With kind regards,

Anita Estes

PLOS Genetics

On behalf of:
